# Sea Surface Temperature Analysis for Fengyun-3C Data Using Oriented Elliptic Correlation Scales

**DOI:** 10.3390/s21238067

**Published:** 2021-12-02

**Authors:** Zhihong Liao, Bin Xu, Junxia Gu, Chunxiang Shi

**Affiliations:** National Meteorological Information Center, Beijing 100081, China; gujx@cma.gov.cn (J.G.); shicx@cma.gov.cn (C.S.)

**Keywords:** oriented elliptic correlation scales, Kalman filtering, sea surface temperature, FY-3C VIRR data

## Abstract

Sea surface temperature (SST) is critical for global climate change analysis and research. In this study, we used visible and infrared scanning radiometer (VIRR) sea surface temperature (SST) data from the Fengyun-3C (FY-3C) satellite for SST analysis, and applied the Kalman filtering methods with oriented elliptic correlation scales to construct SST fields. Firstly, the model for the oriented elliptic correlation scale was established for SST analysis. Secondly, observation errors from each type of SST data source were estimated using the optimal matched datasets, and background field errors were calculated using the model of oriented elliptic correlation scale. Finally, the blended SST analysis product was obtained using the Kalman filtering method, then the SST fields using the optimum interpolation (OI) method were chosen for comparison to validate results. The quality analysis for 2016 revealed that the Kalman analysis with a root-mean-square error (RMSE) of 0.3243 °C had better performance than did the OI analysis with a RMSE of 0.3911 °C, which was closer to the OISST product RMSE of 0.2897 °C. The results demonstrated that the Kalman filtering method with dynamic observation error and background error estimation was significantly superior to the OI method in SST analysis for FY-3C SST data.

## 1. Introduction

Sea surface temperature (SST) is not only an essential parameter in physical oceanography, but also has a substantial impact on the response and drive of globally changing ocean processes [1,2,3,4,5]. At present, a complete high-accuracy SST field is usually obtained by blending in situ measurements and SSTs retrieved from satellite remote sensing data [6,7,8,9]. The analysis of multi-source SST data requires us to consider the effective range of the SST correlation areas and the weight assigned to each SST observation. The effective range of the correlation areas determines the number of effective observations, which mainly depends on the correlation scales applied for SST analysis. The correlation of SST increments is often affected by factors, such as surface ocean currents and the heat transfer effect of SST, and these factors should have orientational variability in different sea areas [10,11,12]. It is generally believed that the oriented ellipse is more suitable as the correlation scale in blended SST analysis, such that the orientation of the correlation scale in different sea areas can be simulated [13,14]. Additionally, the weights of SST observations are determined by objective analysis and error estimation of SST data, and the optimum interpolation (OI) and Kalman filtering methods are commonly used for the objective analysis of SST data [8,15]. OI analysis has the advantages of easy implementation and fast calculation, and OI has been widely applied in fields, such as SST analysis [16,17,18,19]. However, the OI parameters only reflect the spatial variations of the errors from the observation and background fields, and these values are temporally fixed, whereas in theory, the response functions of sensors to SST related to these errors usually change with time [20,21]. Kalman filtering is a classical data assimilation method that is similar to the OI method [22,23], but this technique can separate the background field errors from the observation field errors during data analysis, and the errors from these two fields can constantly be updated in the analysis process. Therefore, the Kalman filtering method can solve the problem that occurs because the error ratio of the observation and background fields cannot be adjusted using the OI method.

In this study, the oriented ellipse as a correlation scale for SST was adopted, and a blended analysis solution using the VIRR SST from the Fengyun-3C (FY-3C) satellite and in situ SST data with the Kalman filtering method was explored by effectively estimating the errors in the observation and background fields. Moreover, the blended SST results were validated and compared with those obtained using the OI method.

## 2. Materials and Methods

### 2.1. Data

In this study, we used the VIRR SST data provided by the National Satellite Meteorological Center (NSMC) of the China Meteorological Administration, which are a Level-2 SST product containing the daytime (VIRRD) and nighttime (VIRRN) values with a spatial grid resolution of 5 km [24,25,26]. This SST product has been available from the Fengyun Satellite Data Center since May 2014 (http://satellite.nsmc.org.cn/PortalSite/Data/Satellite.aspx, accessed on 15 March 2021). Both the VIRRD and VIRRN data were processed by bias correction based on the piecewise regression method before SST analysis [27]. The in situ SSTs were provided by the in situ SST quality monitor (iQuam) system of the US National Oceanic and Atmospheric Administration (NOAA) Center for Satellite Applications and Research (STAR) (http://www.star.nesdis.noaa.gov/sod/sst/iquam/index.html, accessed on 15 March 2021). This system performs quality identification and control for the in situ SST data provided by the Global Telecommunication System (GTS) through statistical evaluations, and the in situ SST datasets are classified with a quality identification of Levels 1–5 [28,29]. The in situ SST data used in this study are the highest-quality buoy data from the iQuam system. Moreover, the OISST product was selected as the SST reference in this study, which is a Level-4 Group for High Resolution Sea Surface Temperature product provided by the US National Climatic Data Center (NCDC) (ftp://data.nodc.noaa.gov/pub/data.nodc/ghrsst/GDS1/L4/GLOB/NCDC/, accessed on 15 March 2021), and this is a daily SST analysis product obtained by OI analysis of the globally distributed Advanced Very High Resolution Radiometer (AVHRR) data and in situ SSTs, with a spatial grid resolution of 0.25° [8,30]. Because of the high data quality, the OISST product is currently one of the most commonly used reference datasets for validating SST products.

### 2.2. Methods

#### 2.2.1. Oriented Elliptic Correlation Scale

In the blended SST analysis, the magnitude of the correlation scale determines the correlation range of SST increments in the target region, thereby affecting the number of valid observations incorporated in the analysis process. In our experiment, an oriented ellipse was used as the basic shape unit of the correlation scale, and it has been proved that the oriented elliptic correlation scales are more effective than the rectangular correlation scales when we apply the OI analysis for Fengyun-3C data [13]. Therefore, the same model for the relationship between correlation and distance was established in this study as follows [13,14]:(1)$$F_{d,\theta }=\exp[\frac{-d}{D\left(\theta \right)}].$$

$F_{s,\theta }$ is the correlation function for distance d and azimuth angle θ in polar coordinates. D(θ) represents the equation of an ellipse whose distance varies with θ:(2)$$D\left(\theta \right)=\frac{L_{max}L_{min}}{\sqrt{L_{max}^{2}\sin^{2}\left(\theta -\varphi \right)+L_{min}^{2}\cos^{2}\left(\theta -\varphi \right)}}$$

$L_{max}$, $L_{min}$, and φ denote the semi-major axis, semi-minor axis, and rotation angle of the ellipse, respectively. These three parameters determine the effective range and direction of the target region, and they have been estimated by using an entire year of SST datasets with the Gauss–Newton iteration method in Liao et al. [13], which were obtained by OI analysis of the VIRRD, VIRRN, and in situ SST data from 2015. The global distributions of the semi-major axis $L_{max}$, semi-minor axis $L_{min}$, and rotation angle φ of the ellipse are illustrated in Figure 1, Figure 2 and Figure 3, respectively.

#### 2.2.2. Estimation of SST Observation Error

Using the Kalman filtering method to blend the multi-source SST data, the observation errors in each data type must be estimated before SST analysis. This study involved three types of observation data: VIRRD, VIRRN, and in situ SSTs. Among them, the in situ SSTs are the highest-quality data processed by the iQuam system, qualified for applications with high quality and accuracy. However, because the daily in situ SST observations were few and sparsely distributed, it was impossible to obtain sufficient samples to estimate the observation error in each global grid space. Thus, a fixed value of 0.1 °C was selected as the error of the in situ SSTs after calculating the standard deviation of the in situ SST increment field in this study. 

The SST observation errors of VIRRD and VIRRN were obtained by calculating the error standard deviation estimation $SD_{est}$ from the optimal matched datasets ($MD_{optimal}$), as shown in the schematic diagram of the SST observation error estimation (Figure 4). To obtain the $MD_{optimal}$ for each SST pixel, the local matchup samples $\left(MD_{local}\right)$ were selected from the matchup datasets of in situ SST, original VIRR SST $\left(T_{s,i,j}\right)$, climatology SST $\left(T_{c,i,j}\right)$ and view zenith angle ($\theta_{i,j}$) within the local window length *L* during a 15 d period prior to the target day, and $T_{c,i,j}$ is the daily SST climatology obtained from 30 years (1982–2011) of OISST data [6]. The initial *L* is 10° for each pixel in the position of the SST matrix (i,j), whereas the value of *L* will be increased by 2.5° each time the number of the local matchup samples $\left(N_{local}\right)$ is less than 300. Then, the estimated standard deviation $\rho_{i,j}$ can be calculated using the following equations [27,31]:(3)$$\{\begin{array}{c} \rho ={\left[{\left(R-\langle R\rangle \right)}^{T}D_{local}^{-1}\left(R-\langle R\rangle \right)\right]}^{0.5}\\ D_{local}=\langle (R-\langle R\rangle ){\left(R-\langle R\rangle \right)}^{T}\rangle \\ R=\left[T_{s},T_{s}-T_{c},\left(T_{s}-T_{c}\right)\left(sec\theta -1\right)\right] \end{array}$$

Here, ***R*** is a vector of regressors obtained from $MD_{local}$, 〈·〉,which denotes averaging of the vector, superscript T stands for the transpose of a vector, and $D_{local}$ is a covariance matrix of regressors. After calculating the parameter of $\rho_{i,j}$ with Equation (3),$\;\left\{\rho_{local}\right\}$ is the collection of $\rho_{i,j}$ for each pixel in the local window that can be obtained, and the $MD_{optimal}$ can be chosen when the difference between the centered estimated standard deviation $\rho_{i,j}$ and $\left\{\rho_{local}\right\}$ is smaller than a finite value S. The initial S is 0.5, and it increases by 0.1 until the number of optimal matchup samples $\left(N_{optimal}\right)$ is larger than 100. Finally, the error standard deviation estimation, $SD_{est}$, of VIRRD and VIRRN can be obtained using the matched datasets of in situ SST and $T_{s,i,j}$ from $MD_{optimal}$.

However, in situ SSTs in high-latitude regions are scarce, especially in polar regions covered by sea ice year-round, where it cannot be provided for bias correction or data validation. In this study, we only selected sea areas between 60° S and 60° N for the SST analysis. The spatial distributions of the SST observation field error were combined with VIRRD, VIRRN, and in situ SSTs on 1 May and 1 October 2016 (Figure 5).

#### 2.2.3. Estimation of SST Background Error 

After obtaining the SST observation field errors, the background field errors in the current iteration should also be estimated for the SST analysis using the Kalman filtering method. According to the Kalman filtering equations, the analysis field error $\sigma_{k,t}^{a}$ in the current iteration *t* can be used to estimate the background field errors $\sigma_{k,t+1}^{b}$ in the next iteration (*t* + 1). However, the practical application of the equations revealed that the background field error grew at a certain ratio during continuous updating, eventually leading to a severe unrealistic deviation of the background field error from the normal range. Therefore, a proportional relationship exists between the estimated analysis field error in the current iteration and the background field error in the next iteration.
(4)$$\sigma_{k,t+1}^{b}=\kappa \sigma_{k,t}^{a}$$
where k denotes the target position and κ is the coefficient of proportionality between the two errors. To select an appropriate κ value, we assume that the SST background field error in the current iteration comes from the observation field error in the previous iteration, and the average of the current observation field error is equal to the average of the background field error in the next iteration, that is, $\bar{\sigma_{t+1}^{b}}=\bar{\sigma_{t}^{o}}$. In this case, the magnitude of the κ value varies with the iteration time. It can be expressed as the ratio of the average error of the current observation field $\bar{\sigma_{t}^{o}}$ to the average error of the current SST analysis field $\bar{\sigma_{t}^{a}}$:(5)$$\kappa =\frac{\bar{\sigma_{t}^{o}}}{\bar{\sigma_{t}^{a}}},$$
and the error of the current analysis field is as follows:(6)$$\sigma_{k,t}^{a}={\left|{\left(\sigma_{k,t}^{b}\right)}^{2}-\sum_{i=1}^{N}w_{ik}\sigma_{ik}^{b}\right|}^{0.5},$$
where $w_{ik}$ is the weight of the pair of observations at position i obtained during the SST analysis and target position k, and $\sigma_{ik}^{b}\;$ is the covariance of errors at the two adjacent positions i and k, which can be estimated using the oriented elliptic error correlation scale model in Equation (1). Based on the above equations, the error standard deviation of the SST background field can be estimated, and then the background field error for each iteration in the SST analysis can be obtained using the Kalman filtering method. It is worth noting that we used the absolute error of the OISST and SST background fields as the error of the background field in the initial iteration. The spatial distributions of the background field error on 1 May and 1 October 2016, are shown in Figure 6.

## 3. Results

Using the errors of SST observation and the background field, the complete blended SST analysis results of 2016 can be obtained using the Kalman filtering method. To analyze the quality of these blended SST products, we selected the OISST products and the blended SST results from the OI method for comparison, and selected 10% of in situ SST data as the independent SST samples for evaluation, which were not used in the SST analysis process. It should be noted that the OI analysis used the same SST observations and oriented elliptic correlation scale model as the Kalman filtering analysis in the study; the values of the noise-to-signal standard deviation ratios for OI analysis were provided by Liao et al. [13], which can be regarded as the error ratios of the observation field to the background field. Figure 7 shows the number of independent SST samples in the global ocean within 10° × 10°grids. Additionally, we selected the daily SST analysis results of 1 May and 1 October 2016, to validate the quality of the analysis product in this study.

### 3.1. Comparison of 2016 SST Analysis Results

Figure 8 shows the distributions of the average bias (BIAS) and error standard deviation (SD) of each SST analysis product estimated using the 2016 independent SST samples in the global 10° × 10° grid space. For the global distribution of BIAS, the values of the OI, Kalman, and OISST results were all approximately 0 °C, indicating that the BIAS values of these three SST results were minimal, and BIAS in the SST field was effectively rejected. The SD values of the OI and Kalman results were smaller than those of the OISST results in low-latitude regions. In contrast, the OI and Kalman results had relatively high SD values in mid- and high-latitude areas, such as the Northwest Pacific, Northwest Atlantic, and South Indian Ocean. Overall, compared to the OI results, the BIAS and SD values of the Kalman results were relatively smaller and much closer to the results of the OISST products. This demonstrated that the Kalman filtering method provides higher blending analysis quality than the OI method, which can effectively reduce the errors from the observation data in the SST analysis.

Figure 9 shows the scatter plots for the comparison of in situ SST with the OI, Kalman, and OISST results for 2016. It can be seen from the scatter plots that the matching points of the three SST analysis results with the independent in situ SST sample were well distributed on the 1:1 diagonal. Although some matching points for the OI and Kalman results deviated from the diagonal, the differences among the three were minimal. This phenomenon indicated that the BIAS in the OI and Kalman results was relatively small, and was similar to the magnitude of that in the OISST product. These analyses results were generally consistent with the comparison of the spatial distributions of their BIAS values in Figure 8.

To further analyze the performance of the products from the OI and Kalman filtering methods, we used the root-mean-square error (RMSE), error standard deviation (SD), correlation coefficient (*R*), and signal-to-noise ratio (SNR) for evaluation. The results in Figure 10 show that the SST fields using the Kalman filtering method had higher SST data quality than the OI results, but the time series variations in each error metric of the Kalman results were still slightly inferior to those of the OISST results, although their values were very close to each other. The statistical results for 2016 in Table 1 reveal that the average RMSE values of the OI and Kalman results were 0.3911 and 0.3243 °C, respectively, compared with 0.2897 °C for the OISST results. These statistical results showed that the SST analysis field obtained using the Kalman filtering method was significantly better than that produced by the OI method with the same data source. This also indicated that the Kalman filtering method with dynamic observation error and background error estimation can provide better analysis results than the OI method only using fixed values of error ratios of each type of SST observation to the background field.

### 3.2. Validation of Two Sets of Daily SST Analysis Results

To directly validate the daily SST analysis results of the Kalman filtering method, we selected two sets of result data from 1 May and 1 October 2016, for quality analysis. The two sets of SST analysis results obtained by the Kalman filtering method with oriented elliptic correlation scales are illustrated in Figure 11. These results showed that both SST fields have reasonable data distributions and can accurately describe the changes in SST in different seasons. This indicated that the Kalman filtering method with oriented elliptic correlation scales can effectively realize the blended analysis of SST data.

Figure 12 shows the difference distributions of the OI and Kalman results with the OISST products from 1 May to 1 October 2016. The OISST products served as the SST reference, and the absolute differences between the two types of SST products and the OISST were plotted in three intervals: 0–0.4, 0.4–0.8, and >0.8 °C. The absolute differences between each SST and the OISST product were mainly distributed in the intervals of 0–0.4 and 0.4–0.8 °C, which is consistent with the differences of 0.4–0.8 °C among the available SST analysis products [16]. This indicated that the OI and Kalman filtering methods using oriented elliptic correlation scales are very effective for blending SST of FY-3C VIRR data. However, the range of difference distributions in the OI results in the interval of 0.4–0.8 °C were larger than those in the Kalman results, suggesting that the SST results obtained using the Kalman filtering method were closer to the OISST data than those obtained using the OI method.

In addition, the Taylor diagrams using the correlation coefficient (R), SST standard deviation (SST SD), and unbiased root-mean-square error (ubRMSE) of each SST are shown in Figure 13. The locations of the in situ SST data, OI and Kalman results, and OISST products nearly overlapped in the Taylor diagrams, and the statistical results in Table 2 show that the SST SD, *R*, and ubRMSE of OI and Kalman results, and OISST products were very close to those of the in situ SST data, and the error indices for the October 1 results obtained using the Kalman filtering method were superior to those of the OISST product. This also indicated that the SST analysis using the Kalman filtering method with oriented elliptic correlation scales was very effective for the VIRR SST data. 

## 4. Discussion

In SST analysis with Kalman filtering, the effective error estimations of the background and observation fields are the keys to obtaining high-quality reconstruction results. Only if correct estimations of these two errors are achieved can we assign appropriate weights to different SST observations during data analysis. The errors of the background field are from the analysis field, which are affected by the SST analysis method and the quality of the SST observations. Therefore, to obtain a high-quality SST analysis product, we should minimize the errors for the SST observations and the errors in SST analysis methods, which are the two main error sources of SST products that should be discussed in this study.

### 4.1. Error Analysis for SST Analysis Methods

Because the OI method uses the same data source and oriented elliptic correlation scale model as the Kalman filtering method in SST analysis, we can assume that the differences between the OI and Kalman results are mainly caused by the errors from the SST analysis methods. It was clarified in Section 3 that the Kalman filtering method, which uses dynamically updated observation field and background field errors, is significantly better than the OI method with the fixed error ratios of the observation field to the background field. These results also showed that different SST analysis methods have a significant effect on the quality of SST products.

The SST results are affected by the SST analysis method that makes the rule of weight assignment, which is based on the error estimation of each input datum. Therefore, the key to effectively minimizing the error of the SST analysis method is to accurately estimate the error of the background field and each observation datum in each time period, such that their weights can be reasonably assigned. The Kalman filtering method in this study selects the $MD_{optimal}$ by analyzing the error of correlation variables of climatology SST and view zenith angle for VIRR SST, and calculates the error standard deviation of the VIRR data with respect to the in situ SST with the $MD_{optimal}$. Because the products of daily SST analysis use the previous analysis results as the current background field, the errors of the background field are estimated by calculating the errors of the SST analysis results based on the oriented elliptic correlation scale model, which are dynamically updated with the iteration in time. However, the OI method uses the fixed error ratios of the observation field to the background field to sign the weights for each input data, which makes it difficult to accurately reflect the error variation of the observation and background fields in practical applications. Therefore, the accuracy of the SST products using the OI method in this study was significantly lower than that of the Kalman filtering method.

### 4.2. Error Analysis for SST Observations

Three types of observation data, including the VIRRD, VIRRN, and in situ SST, were used in this study. Although the in situ SST with the highest quality obtained from the iQuam system was selected as a reference data source for bias correction and validation, the quantity of in situ SST was very low compared to the VIRR SST, which was less than 1% in the SST analysis. Thus, the quality of VIRRD and VIRRN had a significant impact on the SST analysis product. The VIRR data are satellite-retrieved SST products, and they are often affected by several factors, such as anomalous atmospheric conditions, instrument calibration problems, and cloud detection failures, which may produce many biases in SSTs. The multichannel SST algorithm (MCSST) is an operational SST retrieval algorithm for VIRR products [25,32,33], which converts the skin temperature information from VIRR to the bulk temperature as in situ SST using the optimal regression coefficients determined by the least squares method. However, the MCSST is not a function of time or space and highly depends on the relationship between brightness temperatures and in situ SSTs, and the errors from the SST retrieval algorithm will increase with time. 

The errors for VIRRD and VIRRN from the SST retrieval algorithm can be considered systematic biases, which can be eliminated by bias correction using the in situ data in adjacent space and time. As shown in Figure 6, the systematic bias of each SST analysis results in the spatial distribution of BIAS that has been effectively removed, but there are still some large values of SD in many regions, especially in the Kalman results and OISST projection, which can be considered to be mainly caused by the quality of satellite-retrieved SST products. Because the SST errors in VIRR products are relatively large and the accuracy of satellite-retrieved SST products is low compared to the AVHRR products [34,35], the quality of SST analysis results using VIRR data in both the Kalman filtering method and OI method are inferior to those of the OISST product using AVHRR data.

## 5. Conclusions

In this study, the observation errors of VIRR SST were obtained from the bias correction process with optimal matched datasets. The background field errors were estimated using an oriented elliptic correlation scale, and the blended analysis of VIRR SST data and in situ observations was achieved using the Kalman filtering methods. Compared to the OI results with the same SST observations in 2016, the error statistics of the Kalman results were superior to those of the OI results, indicating that the Kalman filtering method with dynamic error estimation had advantages over the OI method using fixed errors for observations and backgrounds. Additionally, the comparison of two separate sets of daily SST analysis results revealed that both the Kalman and OI results matched well with the in situ SST data, but the statistics of the Kalman results were better than those of the OI results, and were relatively closer to the OISST data.

Furthermore, comparative analysis of multiple SST results showed that both the OI and Kalman results in the experiment exhibited good product quality, with average RMSE values of 0.3911 and 0.3243 °C, respectively; however, these values were still higher than the 0.2897 °C from the OISST product. This was mainly because the SST data used in this study were retrieved from the VIRR, which still contains significant data quality defects compared with the AVHRR SST used in the OISST results. Thus, the blended SST analysis products have lower signal-to-noise ratios than the OISST product. Therefore, optimizing the quality of SST retrieval from VIRR is crucial to further improve the accuracy of the multi-source blended SST analysis product based on the Fengyun satellite data.

## Figures and Tables

**Figure 1 sensors-21-08067-f001:**
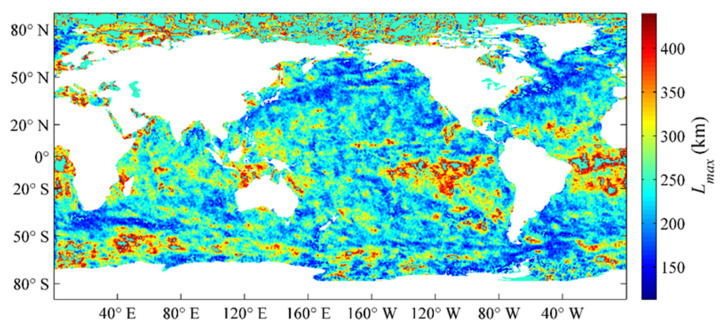
Global distribution of the $L_{max}$ parameter estimated using the optimum interpolation (OI) analysis results of 2015 [13].

**Figure 2 sensors-21-08067-f002:**
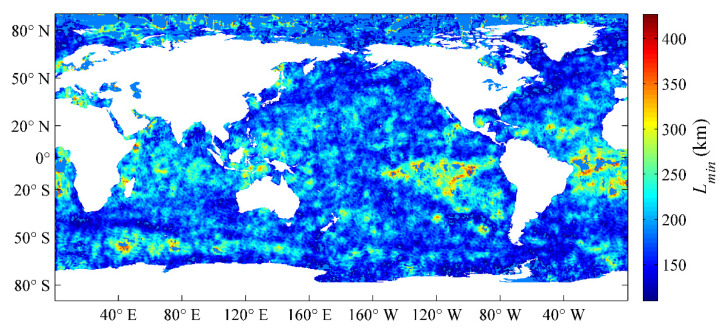
Global distribution of the $L_{min}$ parameter estimated using the optimum interpolation (OI) analysis results of 2015 [13].

**Figure 3 sensors-21-08067-f003:**
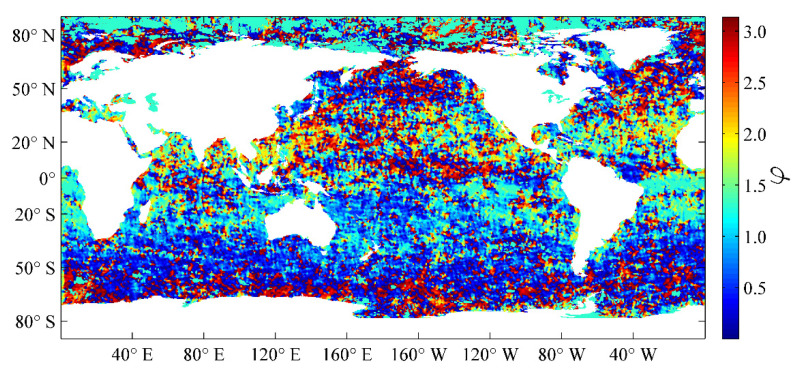
Global distribution of the φ parameter estimated using the optimum interpolation (OI) analysis results of 2015 [13].

**Figure 4 sensors-21-08067-f004:**
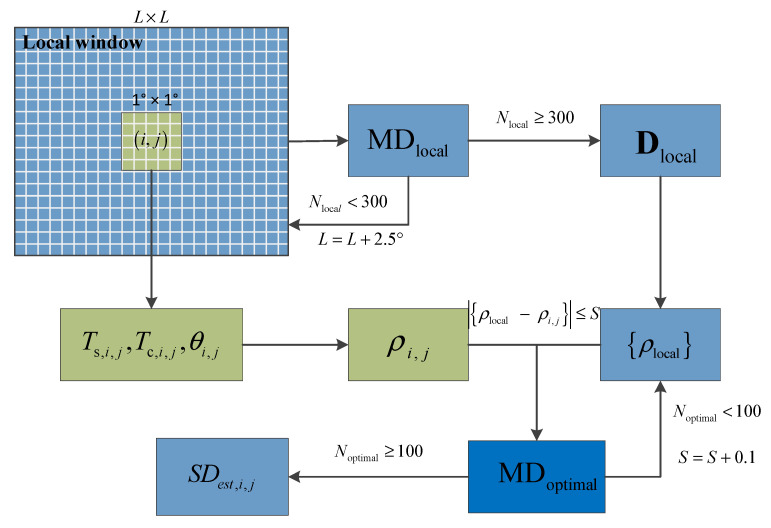
Schematic diagram of the error standard deviation estimation $SD_{est}$ using the optimal matched datasets ($MD_{optimal}$) for visible and infrared radiometer (VIRR) surface sea temperature (SST) products. $T_{s,i,j}$,${\;T}_{c,i,j}$, $\theta_{i,j}$, and $\rho_{i,j}$ separately stand for the original VIRR SST, climatology SST, view zenith angle, and estimated standard deviation in the position of SST matrix (i,j), respectively.

**Figure 5 sensors-21-08067-f005:**
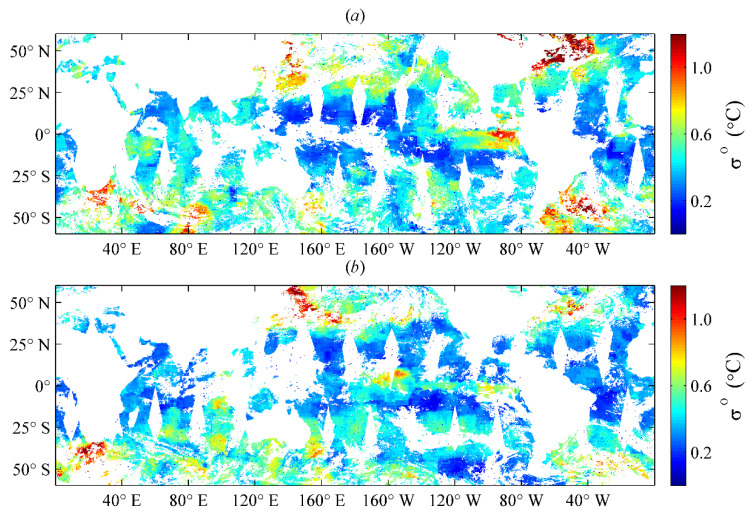
Distributions of SST observation field error on (**a**) 1 May 2016, and (**b**) 1 October 2016.

**Figure 6 sensors-21-08067-f006:**
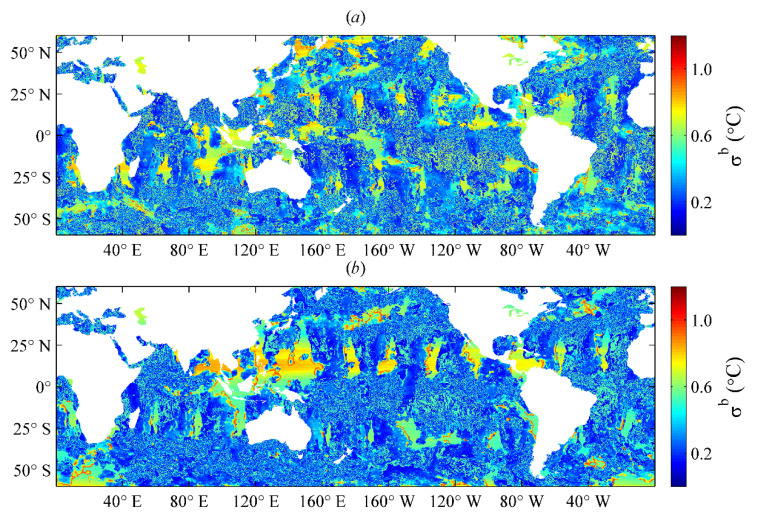
Distributions of SST background error on (**a**) 1 May 2016, and (**b**) 1 October 2016.

**Figure 7 sensors-21-08067-f007:**
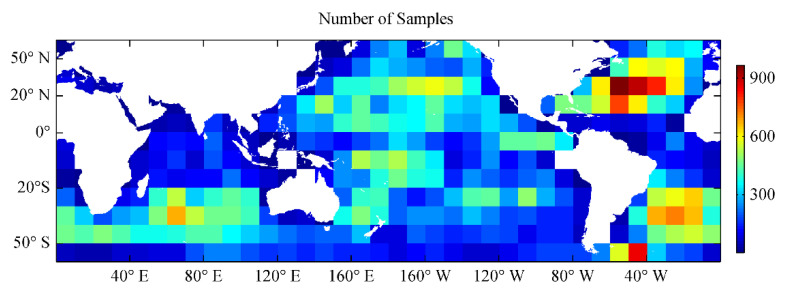
Number of independent SST samples from 2016 in the global ocean within 10° × 10°grids.

**Figure 8 sensors-21-08067-f008:**
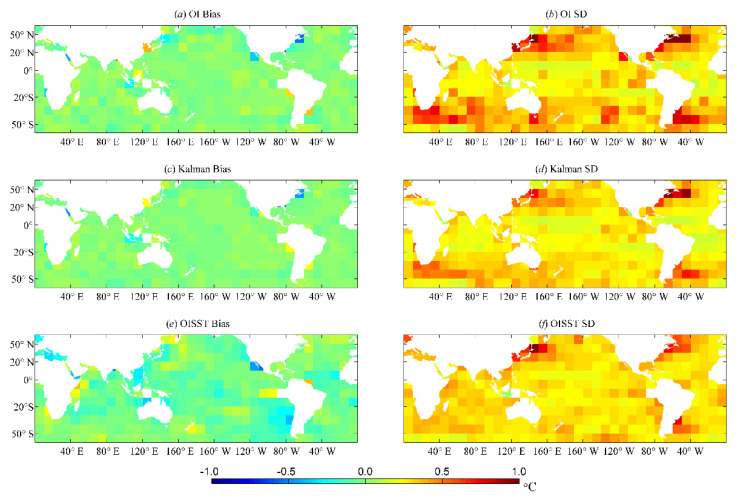
Comparison of the average bias (BIAS) and standard deviation (SD) from 2016 from (**a**,**b**) OI, (**c**,**d**) Kalman and (**e**,**f**) OISST results in global 10° × 10° grids.

**Figure 9 sensors-21-08067-f009:**
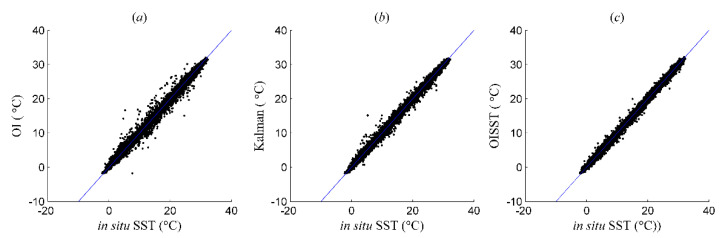
Scatter plots for 2016 from (**a**) OI, (**b**) Kalman and (**c**) OISST results with in situ SSTs.

**Figure 10 sensors-21-08067-f010:**
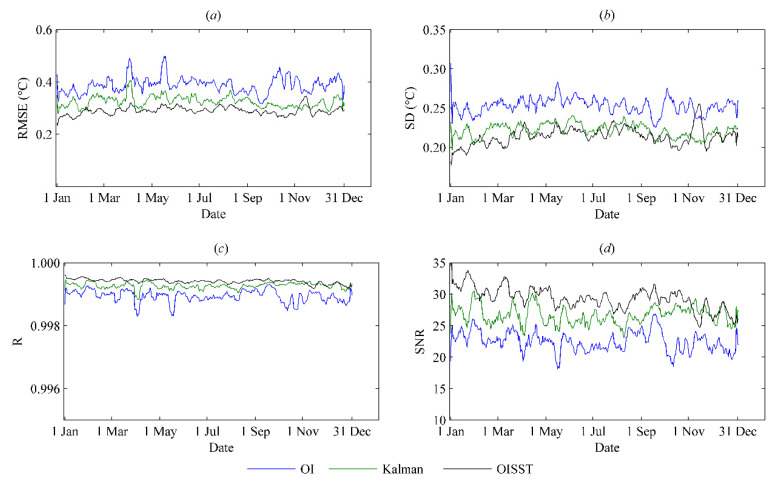
Time series of (**a**) root-mean-square error (RMSE), (**b**) standard deviation (SD), (**c**) correlation coefficient (R), and (**d**) signal-to-noise ratio (SNR) from OI, Kalman and OISST results for 2016.

**Figure 11 sensors-21-08067-f011:**
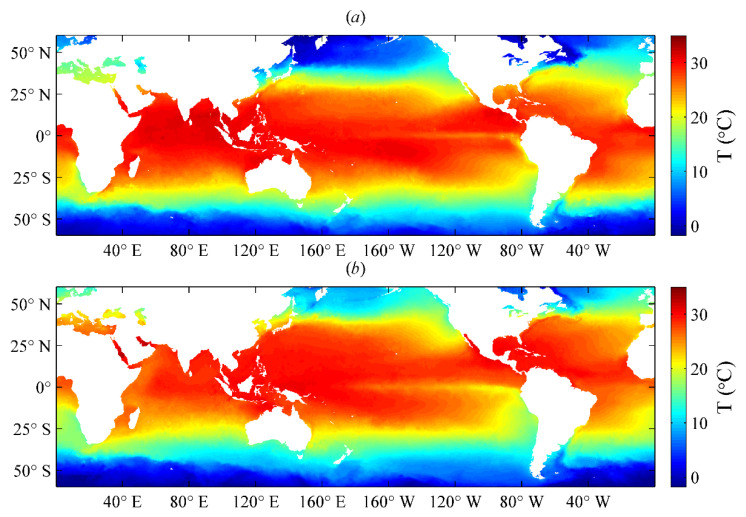
Daily sea surface temperature (SST) analysis results from (**a**) 1 May 2016, and (**b**) 1 October 2016, obtained by the Kalman filtering method with oriented elliptic correlation scales.

**Figure 12 sensors-21-08067-f012:**
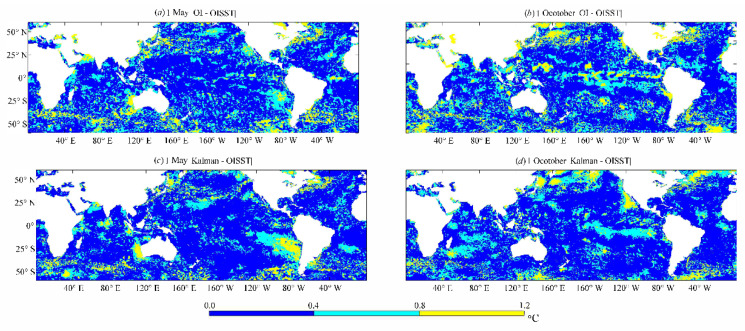
Difference distributions of the OI results (**a**,**c**) and Kalman results (**b**,**d**) with the OISST products from 1 May and 1 October 2016 respectively.

**Figure 13 sensors-21-08067-f013:**
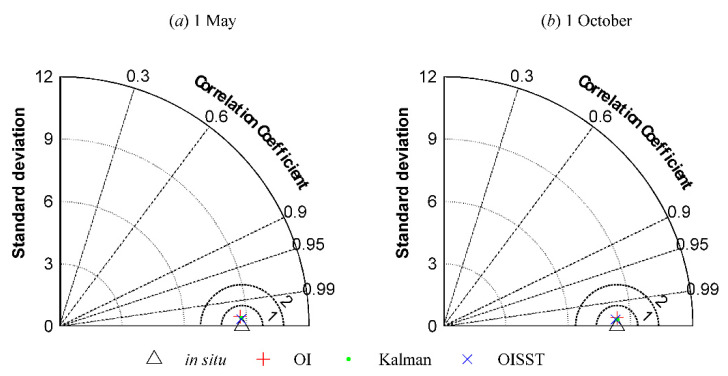
Comparison of the OI, Kalman, and OISST results with the in situ SSTs from (**a**) 1 May 2016 and (**b**) 1 October 2016 using Taylor diagrams.

**Table 1 sensors-21-08067-t001:** Error statistics for OI, Kalman and OISST results for 2016.

	RMSE (°C)	SD (°C)	*R*	SNR	NUM
OI	0.3911	0.2539	0.9989	22.41	82,441
Kalman	0.3243	0.2214	0.9993	26.64	82,441
OISST	0.2897	0.2140	0.9994	29.31	82,441

**Table 2 sensors-21-08067-t002:** Error Statistics of the in situ SST, OI, Kalman, and OISST results from 1 May and 1 October 2016.

	1 May				1 October			
	*R*	SST SD	ubRMSE	NUM	*R*	SST SD	ubRMSE	NUM
In situ	1	8.7775	0	268	1	8.3529	0	260
OI	0.9985	8.7005	0.4779	268	0.9989	8.3659	0.3986	260
Kalman	0.9990	8.7576	0.4001	268	0.9993	8.3719	0.3207	260
OISST	0.9992	8.7452	0.3413	268	0.9992	8.2884	0.3302	260

## Data Availability

Not applicable.

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
