# Peer review of "Sea Surface Temperature Analysis for Fengyun-3C Data Using Oriented Elliptic Correlation Scales"

_sensors, 2021, doi:10.3390/s21238067_

Round 1
Reviewer 1 Report
The authors conducted a very useful and meaningful work. I really appreciate their effort for making the analysis, in particular, using a recent satellite. The results are well presented and the work can be informative to the community. I'd like to recommend the paper to be published with few points being improved.
- There are some scattering points of semi-major axis in the tropical Pacific and Atlantic in Figure 1. Do you have any explanation for them? Though these points can be filtered by the applied method, the filled data quality need justification.
- The comparison between Kalman and OISST shows the OISST is still better performing. Is this because the spatial sampling of in-situ observations are mostly locating in the Atlantic Ocean, but the data of Fengyun-3C is optimized mostly by water samples from Pacific Ocean? Some explanation can be helpful for the reader to have a neutral understanding of the dataset.
- Lastly, instead of making comparison with in-situ observations, can you compare the data with MODIS and other products that has been widely adopted. This can help the community to increase the confidence to use the data.
Author Response
The authors conducted a very useful and meaningful work. I really appreciate their effort for making the analysis, in particular, using a recent satellite. The results are well presented and the work can be informative to the community. I'd like to recommend the paper to be published with few points being improved.
Figure 1 shows the global distributions of the semi-major axis L_max for oriented elliptic correlation scale, which was obtained by using an entire year of SST datasets with the Gauss–Newton iteration method, reflecting the farthest distant related to the location (i, j). When some large values appear in some locations, it indicates that they have a large correlation with their distant regions. And this relationship is closely related to the spatial distribution of sea surface temperatures due to current movements in the oceans, such as the eastern regions of the Pacific and Atlantic Oceans at the equator, where there are large values of correlation scales due to the presence of significant current movements throughout the year. Likewise, some scattering points presented in Figure 1 also are due to the pronounced local current motion in the regions. Unfortunately, the mechanism for the relationship between the current motion and the correlation scale are still unclear, so we cannot further discuss them in this manuscript, and this part of work will be analyzed in the next step research.
2 The comparison between Kalman and OISST shows the OISST is still better performing. Is this because the spatial sampling of in-situ observations are mostly locating in the Atlantic Ocean, but the data of Fengyun-3C is optimized mostly by water samples from Pacific Ocean? Some explanation can be helpful for the reader to have a neutral understanding of the dataset.
It is true that the SST analysis results using Kalman filtering method are still not as good as the OISST, but it is not because the spatial distributions of in-situ observations are different between Atlantic Ocean and Pacific Ocean.
Generally, in order to fully take advantage of in situ data to improve the quality of SST analysis products, most of products including the OISST and the Kalman results in the study use the globally distributed in-situ observations for bias correction and SST analysis. And the in situ observations for evaluations are also globally distributed as shown in Figure 7. We do not select regional in situ observations for data optimization. Therefore, the difference between Kalman and OISST are mainly caused by the different satellite data used in SST analysis.
Meanwhile, we have discussed this problem in Section 4, the analysis errors are divided into two parts: Error Analysis from SST Analysis Methods and Error Analysis from SST Observations, and the SST errors in VIRR products are relatively large and the accuracy of satellite-retrieved SST products is low compared to the AVHRR products[34, 35], which lead to the quality of SST analysis results using VIRR data in both the Kalman filtering method and OI method are inferior to those of the OISST product using AVHRR data.
- Liao, Z., Q. Dong, and C. Xue, Evaluation of sea surface temperature from FY-3C data. International Journal of Remote Sensing, 2017. 38(17): p. 4954-4973.
- Pryamitsyn, V., et al., Evaluation of the initial NOAA AVHRR GAC SST reanalysis version 2 (RAN2 B01). SPIE Defense + Commercial Sensing. Vol. 11420. 2020: SPIE.
3 Lastly, instead of making comparison with in-situ observations, can you compare the data with MODIS and other products that has been widely adopted. This can help the community to increase the confidence to use the data.
According to the description from Group for High Resolution Sea Surface Temperature (GHRSST, https://www.ghrsst.org/ghrsst-data-services/products/), the SST data in satellite swath coordinates, gridded data, and gap-free gridded products are L2, L3 and L4 product, respectively. And they are defined as below:
- L2P data products provide satellite SST observations together with a measure of uncertainty for each observation in a common GHRSST netCDF format. Auxiliary fields are also provided for each pixel as dynamic flags to filter and help interpret the SST data. These data are ideal for data assimilation systems or as input to analysis systems.
- Gridding a single L2P file produces an “uncollated” L3 file (L3U). Multiple L2P files are gridded to produce either a “collated” L3 file (L3C) from a single sensor or a “super-collated” L3 file from multiple sensors (L3S).
- L4 gridded products are generated by combining complementary satellite and in situ observations within Optimal Interpolation systems. L4 gridded products are provided in GHRSST netCDF format. These data are ideal for model diagnostic studies, model boundary condition specification and model initialisation.
By these definitions, the SST of MODIS can be regarded as the L2 or L3 SST data, as well as the VIRR SST data from Fengyun-3C used in this study, and these kind of SST fields can be obtained in daytime or nighttime that are usually incomplete on the global ocean. While the OISST and the SST results from the OI and Kalman method are the L4 SST products with the completed SST fields, which blend the SST retrieval data from satellite and the in situ observations. Therefore, using the MODIS data for comparison is not suitable in this study because they are totally different.
Moreover, the OISST with the same temporal-spatial resolution to the OI and Kalman results, is the only L4 SST product using one type of satellite data, which is the most appropriate reference data in the experiment, and the OISST product is one of the most commonly used SST datasets for validating. Thus we selected the OISST for comparison in the study. While other famous L4 SST products in GHRSST, such as the OSTIA, have different spatial resolution, and they combine a lot of satellite data including SST from microwave and infrared satellite instruments. Hence, we think they are not suitable to be used for comparison in this study.
Reviewer 2 Report
This is a resubmitted job.
Thanks to the author for correcting the paper according to my previous comments. I have no more comments.
I’m here to give minor’s suggestions, and I hope to see other reviewers’ opinions.
Author Response
Here are the other reviewers’ opinions and the corresponding responses:
The authors conducted a very useful and meaningful work. I really appreciate their effort for making the analysis, in particular, using a recent satellite. The results are well presented and the work can be informative to the community. I'd like to recommend the paper to be published with few points being improved.
Figure 1 shows the global distributions of the semi-major axis L_max for oriented elliptic correlation scale, which was obtained by using an entire year of SST datasets with the Gauss–Newton iteration method, reflecting the farthest distant related to the location (i, j). When some large values appear in some locations, it indicates that they have a large correlation with their distant regions. And this relationship is closely related to the spatial distribution of sea surface temperatures due to current movements in the oceans, such as the eastern regions of the Pacific and Atlantic Oceans at the equator, where there are large values of correlation scales due to the presence of significant current movements throughout the year. Likewise, some scattering points presented in Figure 1 also are due to the pronounced local current motion in the regions. Unfortunately, the mechanism for the relationship between the current motion and the correlation scale are still unclear, so we cannot further discuss them in this manuscript, and this part of work will be analyzed in the next step research.
2 The comparison between Kalman and OISST shows the OISST is still better performing. Is this because the spatial sampling of in-situ observations are mostly locating in the Atlantic Ocean, but the data of Fengyun-3C is optimized mostly by water samples from Pacific Ocean? Some explanation can be helpful for the reader to have a neutral understanding of the dataset.
It is true that the SST analysis results using Kalman filtering method are still not as good as the OISST, but it is not because the spatial distributions of in-situ observations are different between Atlantic Ocean and Pacific Ocean.
Generally, in order to fully take advantage of in situ data to improve the quality of SST analysis products, most of products including the OISST and the Kalman results in the study use the globally distributed in-situ observations for bias correction and SST analysis. And the in situ observations for evaluations are also globally distributed as shown in Figure 7. We do not select regional in situ observations for data optimization. Therefore, the difference between Kalman and OISST are mainly caused by the different satellite data used in SST analysis.
Meanwhile, we have discussed this problem in Section 4, the analysis errors are divided into two parts: Error Analysis from SST Analysis Methods and Error Analysis from SST Observations, and the SST errors in VIRR products are relatively large and the accuracy of satellite-retrieved SST products is low compared to the AVHRR products[34, 35], which lead to the quality of SST analysis results using VIRR data in both the Kalman filtering method and OI method are inferior to those of the OISST product using AVHRR data.
- Liao, Z., Q. Dong, and C. Xue, Evaluation of sea surface temperature from FY-3C data. International Journal of Remote Sensing, 2017. 38(17): p. 4954-4973.
- Pryamitsyn, V., et al., Evaluation of the initial NOAA AVHRR GAC SST reanalysis version 2 (RAN2 B01). SPIE Defense + Commercial Sensing. Vol. 11420. 2020: SPIE.
3 Lastly, instead of making comparison with in-situ observations, can you compare the data with MODIS and other products that has been widely adopted. This can help the community to increase the confidence to use the data.
According to the description from Group for High Resolution Sea Surface Temperature (GHRSST, https://www.ghrsst.org/ghrsst-data-services/products/), the SST data in satellite swath coordinates, gridded data, and gap-free gridded products are L2, L3 and L4 product, respectively. And they are defined as below:
- L2P data products provide satellite SST observations together with a measure of uncertainty for each observation in a common GHRSST netCDF format. Auxiliary fields are also provided for each pixel as dynamic flags to filter and help interpret the SST data. These data are ideal for data assimilation systems or as input to analysis systems.
- Gridding a single L2P file produces an “uncollated” L3 file (L3U). Multiple L2P files are gridded to produce either a “collated” L3 file (L3C) from a single sensor or a “super-collated” L3 file from multiple sensors (L3S).
- L4 gridded products are generated by combining complementary satellite and in situ observations within Optimal Interpolation systems. L4 gridded products are provided in GHRSST netCDF format. These data are ideal for model diagnostic studies, model boundary condition specification and model initialisation.
By these definitions, the SST of MODIS can be regarded as the L2 or L3 SST data, as well as the VIRR SST data from Fengyun-3C used in this study, and these kind of SST fields can be obtained in daytime or nighttime that are usually incomplete on the global ocean. While the OISST and the SST results from the OI and Kalman method are the L4 SST products with the completed SST fields, which blend the SST retrieval data from satellite and the in situ observations. Therefore, using the MODIS data for comparison is not suitable in this study because they are totally different.
Moreover, the OISST with the same temporal-spatial resolution to the OI and Kalman results, is the only L4 SST product using one type of satellite data, which is the most appropriate reference data in the experiment, and the OISST product is one of the most commonly used SST datasets for validating. Thus we selected the OISST for comparison in the study. While other famous L4 SST products in GHRSST, such as the OSTIA, have different spatial resolution, and they combine a lot of satellite data including SST from microwave and infrared satellite instruments. Hence, we think they are not suitable to be used for comparison in this study.
This manuscript is a resubmission of an earlier submission. The following is a list of the peer review reports and author responses from that submission.
Round 1
Reviewer 1 Report
Comments to the Author.
I tried to review this paper carefully but found that it is too difficult to read. Almost every line has some grammar errors. I tried to ignore the errors by reading between lines and found the paper lacks enough novelty to be considered publishable. From the abstract, they present the Kalman filtering method with dynamic observation error. But some parts have been discussed by Liao et al. 2021, the author uses the almost same figure and equations. I read back and forth and tried to give the paper an objective evaluation. However, the presentation quality is so poor so that I cannot do it properly. I suggest the rejection of the paper, and encourage the authors revise it extensively before resubmission to any other journal.
Major Comments.
1. Line 18 and 19: Since both of the Kalman analysis and the OI analysis have the RMSEs of 0.3243 °C, I cannot see any improvement.
2. In this study, in situ SSTs were provided by the in situ SST Quality Monitor (iQuam) which contains the buoys, which temperature is at a depth of about 20 cm.": An important question is if the drifting buoys are able to help in validating satellite SST? Fengyun-3 satellite SST is derived from visible and infrared radiometer (VIRR). The infrared SST is the skin layer SST. Have you made any adjustments of the diurnal warming layer?
3. All of the following comparisons are based on the paper:
Liao, Z., Xu, B., Zhang, L., Gu, J., & Shi, C. (2021). Optimum interpolation analysis for Fengyun-3C sea surface temperature data using oriented elliptic correlation scales. Remote Sensing Letters, 12(6), 585-593.
Figure 1-2 in the manuscript is taken from Figure 4-5 in Liao et al. 2021
The equations 1-2 in the manuscript are taken from Equation 2-3 in Liao et al. 2021.
The Figure 3 in the manuscript are modified from Figure 6 in Liao et al. 2021
Minor Comments.
English editing required
There are lots of errors related with the typo, overall expression in this article seems not to be scientific and professional. The author should define abbreviations and acronyms the first time they are used in the text.
Line 12:
remove one "observation"
Line 13:
"optimal " should be " optimallly"
Line 16:
"comparions " should be "comparisons"
Line 19:
" more closer" should be "closer "
Line 57:
" comparied" should be "compared "
Line 63:
" availabled" should be "available "
Line 88:
"a oriented " should be "an oriented "
Line 98:
" were obtained" should be " was obtained"
Line 149:
" increase by" should be " increases by"
Line 153:
" especilly " should be " especially "
Line 205:
" flitering analysis" should be "filtering analysis "
Line 207:
" which can be regards" should be " which can be regarded"
Line 223:
" comparing to" should be "compared to "
Line 266:
" oberservations" should be " observations"
Line 300:
" were nearly overlap" should be "were nearly overlapped "
Line 321 and 335:
"minmize " should be "minimize "
Line 340:
" respect to" should be " with respect to"
Line 345:
" to asign the" should be " to assign the"
Line 370:
"can considered " should be "can be considered "
Line 377:
" optimal matched" should be " optimally matched "
Line 380:
"obervations of " should be "observations of "
Line 367 and 392:
" SST anlysis " should be " SST analysis "